# Community-wide deworming strategies to reduce high hookworm burden in endemic communities: Results from a cluster randomized trial in Southern India

Rohan Michael Ramesh[1☉], Rajiv Sarkar[1,2☉]*, Vasanthakumar Velusamy[1], Srinivasan Venugopal[1], Anuradha Rose[3], Venkata R. Mohan[3], Vinohar Balraj[3], Vedantam Rajshekhar[4], Kuryan George[3], Jayaprakash Muliyil[1,3], Nicholas C. Grassly[5], Simon J. Brooker[6], Roy M. Anderson[5], Gagandeep Kang[1,7], Sitara Swarna Rao Ajjampur[1]*

1 The Wellcome Trust Research Laboratory, Division of Gastrointestinal Sciences, Christian Medical College, Vellore, Tamil Nadu, India, 2 Indian Institute of Public Health, Shillong, India, 3 Department of Community Health, Christian Medical College, Vellore, Tamil Nadu, India, 4 Department of Neurological Sciences, Christian Medical College, Vellore, Tamil Nadu, India, 5 Department of Infectious Disease Epidemiology, School of Public Health, Faculty of Medicine, Imperial College London, London, United Kingdom, 6 London School of Hygiene and Tropical Medicine, London, United Kingdom, 7 Gates Foundation, Seattle, Washington, United States of America

☉ RR and RS are Joint First Authors
* rajiv.sarkar@iiphs.ac.in (RS); sitararao@cmcvellore.ac.in (SSRA)

## Abstract

### Objectives

Targeted deworming of high-risk groups (usually school-based programs), although effective in controlling morbidity, may not be sufficient to reduce hookworm burden in endemic communities due to the persistence of infection in the adult population. The effectiveness of community-wide mass drug administration (MDA) strategies were evaluated in a hookworm-endemic tribal community in southern India.

### Methods

In this open-label, cluster-randomized community-intervention trial, 45 villages comprising of 2770 households were randomly assigned (1:1:1) to receive community-wide MDA with albendazole at modified intervals: 1) single-cycle, 2) two-cycles (one month apart) and 3) four-cycles (two cycles one month apart, followed by another two after six months). The primary outcome was the prevalence of hookworm infection, and the co-primary outcome was the mean intensity of infection at 12 months post-MDA, assessed through a cross-sectional coprological survey. Secondary outcomes included prevalence and intensity of infection after each MDA cycle and at 3-, 6-, 9- and 24-months post-intervention. Analysis was by intention-to-treat. This trial is registered with the Clinical Trials Registry - India (CTRI/2013/05/003676).

**Data availability statement:** The raw data required to reproduce the findings of the study are available to download from the Open Science Framework through the following URL: https://osf.io/34br6/?view_only=ba891c24459b4023b83d2c1b1347f9f4.".

**Funding:** This study was funded by the Wellcome Trust/DBT India Alliance (https://www.indiaalliance.org/) through an Early Career Fellowship to RS (Grant number: IA/E/12/1/500750). SJB was supported by a Wellcome Trust Senior Fellowship in Basic Biomedical Science (Grant number: 098045). The funders did not play any role in the study design, data collection and analysis, decision to publish, or preparation of the manuscript.

**Competing interests:** The authors have declared that no competing interests exist.

## Results

The prevalence (95% confidence interval) of hookworm infection at baseline was 14.8% (11.1–19.5%), 18.5% (15.4–22.0%) and 22.4% (17.4–28.2%) in the single-, two-, and four-cycle arms, respectively, which reduced to 5% (2.9-8.6%), 4% (1.8-8.4%) and 1.2% (0.3-4.0%) at 12 months post-intervention. At 12 months post-intervention, both prevalence (OR=0.41; $P=0.016$) and intensity (IRR=0.29, $P<0.001$) of infection were significantly lower in the four-cycle arm as compared to the single-cycle arm, but the difference was not statistically significant at 24 months. No statistically-significant difference in prevalence or intensity of infection was observed between single- and two-cycle arms at any time-point.

## Conclusions

Multiple cycles of community-wide MDA may help reduce the burden of hookworm in endemic communities, but improvement in water, sanitation, and hygiene practices are needed for long-term sustainability.

### Author summary

Evidence suggests that community-wide deworming might work better than targeted school-based deworming in reducing hookworm infection rates in endemic populations. We evaluated the effectiveness of modified community-wide deworming strategies in reducing hookworm burden in an endemic community in southern India. Forty-five tribal villages were randomized in a 1:1:1 ratio to receive either a single-cycle, two-cycle (two rounds one month apart), or four-cycle (two rounds one month apart and then another two after six months) of community-wide deworming. The prevalence and intensity of infection were measured at baseline and 3, 6, 9 12 and 24 months after treatment through cross-sectional surveys. The post-treatment prevalence of hookworm infection reduced considerably in all the three treatment arms. Compared to the single-cycle arm, prevalence and intensity of hookworm infection were significantly lower in the four-cycle arm at 3, 6 and 12 months after deworming; prevalence or intensity of infection did not differ significantly between the single- and the two-cycle arms at any time point. The findings of this study suggest that multiple rounds of community-wide deworming may help reduce hookworm burden in endemic communities, but long-term sustainability will require improvement in water, sanitation, and hygiene practices.

## 1. Introduction

Soil transmitted helminths, including roundworm (*Ascaris lumbricoides*), whipworm (*Trichuris trichiura*), and hookworms (*Necator americanus* and *Ancylostoma*

*duodenale*) are a major public health concern in resource-poor settings, with an estimated 642.7 million infections worldwide in 2021 [1]. These infections typically impact impoverished and marginalized communities, particularly those lacking adequate access to clean water, sanitation, and hygiene (WASH), especially in tropical and subtropical regions. They are collectively addressed due to their shared diagnostic procedures and treatment regimens [1]. In India, 27% of the preschool- and school-age children (pre-SAC and SAC) are estimated to be infected with STH [2]. Soil-transmitted helminth infections in children are associated with iron deficiency anemia, poor nutritional status, stunted growth, and reduced physical and cognitive abilities, which may adversely impact school performance and future economic productivity [3,4].

To control STH-associated morbidity, the World Health Organization (WHO) advocates targeted periodic deworming of pre-SAC, SAC and women of child-bearing age living in endemic areas as the mainstay STH-control strategy, along with provision of adequate sanitation, improved hygiene practices and access to safe water [5]. However, even if coverage were optimized, the current STH strategy of targeting children would likely need to be continued indefinitely as prevailing socio-economic and environmental conditions, including poor sanitation, contribute to high re-infection rates in children from untreated reservoirs, particularly for hookworm infections [6,7]. Recent surveys from India have demonstrated a higher burden of hookworm infections in adults, than *Ascaris* and *Trichuris* [8–10]. Findings from large-scale community-intervention trials [11,12], as well as modeling-based estimates [13], suggest that community-wide mass drug administration (MDA) is more effective in reducing hookworm prevalence and/or intensity than targeted school-based deworming. Community-wide MDA programs have also been found to be cost-effective in DALYs averted due to economies of scale [14].

Despite the demonstrable impact of community-wide MDA, there is paucity of studies exploring different doses and regimens to identify the optimal treatment schedule that can significantly reduce hookworm burden in endemic populations. Furthermore, community intervention studies have seldom considered the probability of reinfection occurring shortly after treatment due to the presence of infective larvae in the soil during the extrinsic incubation period [15]. This community-intervention trial evaluates the effectiveness of modified, community-wide MDA strategies that account for the extrinsic incubation period of hookworm larval stages, in reducing the worm burden in a hookworm-endemic tribal community in southern India.

## 2. Methods

**Ethics Statement**: Ethical approval for the study was obtained from the Institutional Review Board (IRB) of Christian Medical College (CMC), Vellore, India. The trial was registered with the Clinical Trial Registry – India (Trial registration number: CTRI/2013/05/003676). The study was initiated after discussions with local community leaders to explain the purpose of the study and procedures. Information sheets in Tamil, the local language, were provided to all prospective participants. Written informed consent was obtained from all participants or their parents/legal guardians (for minors); additionally, children between the ages of 7 and 17 years provided their assent to participate.

### 2.1. Study design and setting

An open-label, cluster-randomized trial was conducted in the tribal block of Jawadhu Hills (JH) in southern India from September 2013 to December 2016. The JH block borders the Vellore and Tiruvannamalai districts of Tamil Nadu and is located on hilly terrain with poor road access, lack of safe drinking water, and poor sanitation facilities [16]. It is predominantly occupied by tribal agrarian communities, many of whom temporarily migrate to coffee plantations in the neighboring states as short-term laborers during non-agricultural seasons. The predominant STH species is hookworm, with the prevalence and intensity of infection increasing with age [10]. From 2007 to 2015, residents aged ≥1 year typically received multiple rounds of annual MDA with diethylcarbamazine (DEC) and albendazole as part of a national program for the elimination of lymphatic filariasis [17].

## 2.2. MDA strategies

The trial design, data collection methods, and follow-up have been presented in detail elsewhere [16]. Briefly, a cluster-randomized trial was conducted in 45 tribal villages, which were randomized into three groups (1:1:1 ratio): (a) annual community-wide MDA (one-cycle group); (b) two rounds of community-wide MDA at one-month intervals (two-cycle group); and (c) four rounds of community-wide MDA – two rounds one month apart at the beginning, followed by another two after six months (four-cycle group). The treatment intervals in the two- and four-cycle groups were based on hookworm biology: the two-cycle group covered the larvae's extrinsic incubation period in the soil to reduce reinfection [15], whereas the four-cycle group was added to account for the survival of infective larvae for several weeks in the soil under favorable environmental conditions [18].

Eligible participants received anthelminthic treatment with albendazole (400 mg) distributed by trained field workers through door-to-door household visits. Individuals with a history of seizure, epilepsy, known neurological disorder, known history of hypersensitivity to albendazole, serological evidence of HIV infection, immune-compromise or currently pregnant were excluded due to concerns around the perceived risks of treatment in these groups. Participants were interviewed the subsequent day to estimate treatment coverage and were followed up for three consecutive days and again 1 week post-treatment to record any adverse/serious adverse events following the intervention. Households that were not available for treatment were revisited the next day. Households were excluded if the dwelling could not be found, was vacant, or had no adult resident at home on three consecutive visits. The participants and trial personnel were not masked to allocation due to the nature of the trial, although the laboratory personnel analyzing the stool samples were unaware of the treatment allocation status.

The treatment (MDA) strategy was at the community-level; however, the study objectives pertain to the individual participant-level. The unit of randomization was a tribal village with 15–100 households. Villages proximal to motorable roads were preferred, anticipating adverse neurological events associated with asymptomatic neurocysticercosis in pork-consuming communities, such as the trial population [19,20]. The study's enrollment started in October 2013 and was completed across all 45 villages by November 2014. The MDA (per protocol) was completed in November 2014 and follow-up data collection for all study villages was completed by November 2016. The 'National Deworming Day' program (bi-annual targeted deworming of PSAC and SAC children between 1 and 19 years of age, held every February and August), which commenced from February 2015 [21], continued concurrently in all study villages towards the end of the follow-up period.

## 2.3. Baseline census

The baseline demographic and socio-economic characteristics of the residents were captured through a door-to-door household census of the study villages [16]. Trained interviewers administered structured questionnaires to all participants (primary caregivers in case of minors) to obtain information on demographic and employment history (including details of agriculture-related activities), footwear usage, defecation and handwashing practices, type of flooring, and animal contact. The respondents' socio-economic status (SES) was calculated using a locally developed and validated SES scale that accounts for caste, type of housing, land ownership, educational and occupational status of the head of the household [22]. Based on the SES score tertile, respondents were classified as belonging to low, medium, or high SES households.

## 2.4. Coprological surveys

Cross-sectional coprological surveys were conducted at baseline, after every MDA cycle, and every three months post-MDA, for one year, and at the end of two years to ascertain the prevalence and intensity of hookworm infection. The follow-up surveys were standardized relative to the timing of the final MDA round within each intervention arm, beginning three months after the single MDA round in one-cycle villages, three months after the second round in two-cycle villages (i.e., four months after baseline), and three months after the fourth round in four-cycle villages (i.e., eight months after baseline) (see Fig 2 of [16]).

Participants for the coprological surveys were randomly selected from among the permanent residents of the study villages aged 2–70 years (both years inclusive) who met the eligibility criteria; exclusions included syndromic or serological evidence of HIV infection or other immunocompromising conditions, pregnancy, a history of seizures, epilepsy or other neurological disorders, and known hypersensitivity to albendazole. The door-to-door baseline census provided the sampling frame. For each survey round, eligible residents were randomly ordered, with the first 30 participants from each village, who provided written informed consent and at least one stool sample, enrolled.

Selected participants provided up to three stool samples, collected in a plastic container labeled with a unique identification number. Individuals who could not provide a sample immediately were contacted again later that day or the next day. Stool samples were stored in a fridge in the field office (4–8 degrees Celsius) and transported to Christian Medical College, Vellore, periodically (2–3 times a week; within 72 hours of collection), based on the number of samples collected. They were then examined microscopically for the presence of hookworm ova using saline and iodine wet preparation [23]. A senior technician re-examined 10% of slides for quality control. Samples positive for hookworm ova were then tested by the McMaster method to measure the number of eggs per gram of stool (EPG) to estimate the intensity of infection [24].

### 2.5. Outcomes

The primary outcome measure was the effectiveness of treatment, evaluated based on the prevalence and intensity of infection at the end of 12 months after the last MDA cycle among all the enrolled individuals. Secondary outcome measures included prevalence and intensity of infection after each treatment cycle at 3-, 6-, 9- and 24-month post-MDA.

### 2.6. Statistical analysis

Data were analyzed using STATA version 14.2 for Windows (StataCorp, College Station, Texas) and R version 3.2.2 (http://www.r-project.org/). All variables were examined using descriptive statistics (measures of central tendency [means and medians], dispersion [standard deviations, interquartile ranges]) for continuous variables, frequency counts, and marginal percentages [with 95% confidence intervals] for categorical variables). The success of randomization was evaluated by comparing the baseline characteristics of the groups without formal statistical testing.

The prevalence and mean intensity of infection were calculated for each village. The age-prevalence and age-intensity profiles of hookworm infection were explored by computing age-specific estimates of hookworm prevalence and intensity. A non-parametric Wilcoxon-type test for trend was used to determine the change in prevalence and intensity of infection with age [25]. Taylor linearized standard errors accounted for the clustered data structure [26]. The baseline demographic and behavioral factors associated with hookworm infection were identified using logistic regression analysis and odds ratios (OR) with 95% confidence intervals (CI) calculated; cluster-robust standard errors accounted for the non-independence of observations [27].

The primary trial analysis was an intention-to-treat analysis comparing the prevalence and intensity of hookworm infection between the three groups at the end of 12 months after the last MDA cycle. Generalized estimating equations (GEE) with a binomial distribution and logit link function for hookworm prevalence and negative binomial distribution with log link function for hookworm intensity were fitted. Exchangeable correlation structure and robust standard errors were used, and age, gender and baseline hookworm prevalence/mean intensity were adjusted for in all models. The effect of clustering at the village and the household level was explored through multilevel (hierarchical) modeling.

## 3. Results

### 3.1. Enumeration of the study villages, enrolment, follow-up, and MDA coverage

The baseline survey identified a total population of 11857 residents in the 45 study villages, with a median (IQR) of 238 (203–289) population per village [16]. The spatial distribution of the study villages, by intervention group, is presented as S1 Fig.

On comparing the socio-demographic profile of the households, collected through the baseline survey, there were more households from the low SES category residing in the single-cycle group (30.6%) compared to the two- (26.6%) and four-cycle groups (25.5%). The proportion of households with at least one member engaged in agriculture-related activities (94.2%), or with a public tap or borewell as the primary source of drinking water (89.4%) in the two-cycle villages was higher compared to the other two groups (Table 1).

Of the 11857 residents, 8681 (73.2%) fulfilled the eligibility criteria and consented to participate in the trial. The primary reason for non-participation was migration outside the study area for temporary employment: 1947/11857 (16.4%) residents were absent in their respective villages during the study period, as illustrated in Fig 1.

The per-protocol MDA coverage (all doses of albendazole received as per the allocated intervention) was highest in the single-cycle group (85.6%, 2458/2872), followed by two- (77%, 2010/2612) and four-cycle (50.4%, 1612/3197) groups. However, the overall MDA coverage (at least one dose received) was slightly higher in the two-cycle group (94.8%, 2475/2612) compared to the four-cycle (93.8%, 2998/3197) group (Fig 1). The individuals who missed MDAs were more likely to be teenagers (15–19 years) or adult males (20–54 years), reflecting the migratory pattern of the study population. These demographic subgroups (teenagers and adult males) were also less likely to participate in the coprological surveys (S2 Fig).

### 3.2. Baseline epidemiology of hookworm infection

A total of 2082 participants (median [IQR] of 46 [44–48] participants per village) provided one or more stool samples; 1727 (83%) provided all three stool samples as per protocol. There was a wide variation in the prevalence of hookworm infection between the villages, ranging from 2.1- 44.2%. The distribution of the intensity profile of hookworm infection showed a highly aggregated behavior: only 1.2% of the sampled population had moderate-to-high intensity infection (MHI, ≥ 2000 EPG). Both the prevalence and intensity of hookworm infection increased with age (S1 Table). Also, males (20.3%, 95% CI: 17.3-23.6%) had a slightly higher prevalence of hookworm infection than females (16.9%, 95% CI: 13.9-20.3%), although the intensity of infection was comparable between both sexes (Mean [SE] EPG = 125 [26] and 126 [28] for males and females, respectively). Analysis of the intensity of infection in hookworm-infected participants revealed that those with all three positive samples had the highest egg counts (mean EPG: 3777, SE: 735), followed by those with two positive samples (mean EPG: 896, SE: 217) and one positive sample (mean EPG: 192, SE: 21), respectively (Kruskal Wallis test, $P < 0.001$). The baseline (pre-intervention) prevalence of hookworm infection was 14.8% in the single-cycle group, 18.5%

**Table 1. Comparison of baseline characteristics of the study villages assigned to different MDA regimens.**

| Baseline characteristics | Single cycle (15 villages) n (%) | Two cycles (15 villages) n (%) | Four cycles (15 villages) n (%) |
|---|---|---|---|
| Total number of households | 944 | 851 | 975 |
| Total number of residents | 3927 | 3705 | 4225 |
| Median (IQR) family size | 5 (4–6) | 5 (4 –6) | 5 (4 –6) |
| Household socio-economic status | | | |
| Low | 289 (30.6%) | 226 (26.6%) | 249 (25.5%) |
| Middle | 440 (46.6%) | 409 (48.1%) | 477 (48.9%) |
| High | 215 (22.8%) | 216 (25.4%) | 249 (25.5%) |
| Proportion of households with at least one member engaged in agriculture-related activities | 872 (92.5%) | 802 (94.2%) | 879 (90.3%) |
| Proportion of households with a functional toilet | 1 (0.1%) | 3 (0.4%) | 3 (0.3%) |
| Proportion of households with public tap or bore-well as the primary source of drinking water | 769 (81.6%) | 761 (89.4%) | 738 (75.8%) |
| Proportion of households reporting having one or more domesticated animal | 772 (81.9%) | 708 (83.2%) | 810 (83.2%) |

PLOS Neglected Tropical Diseases

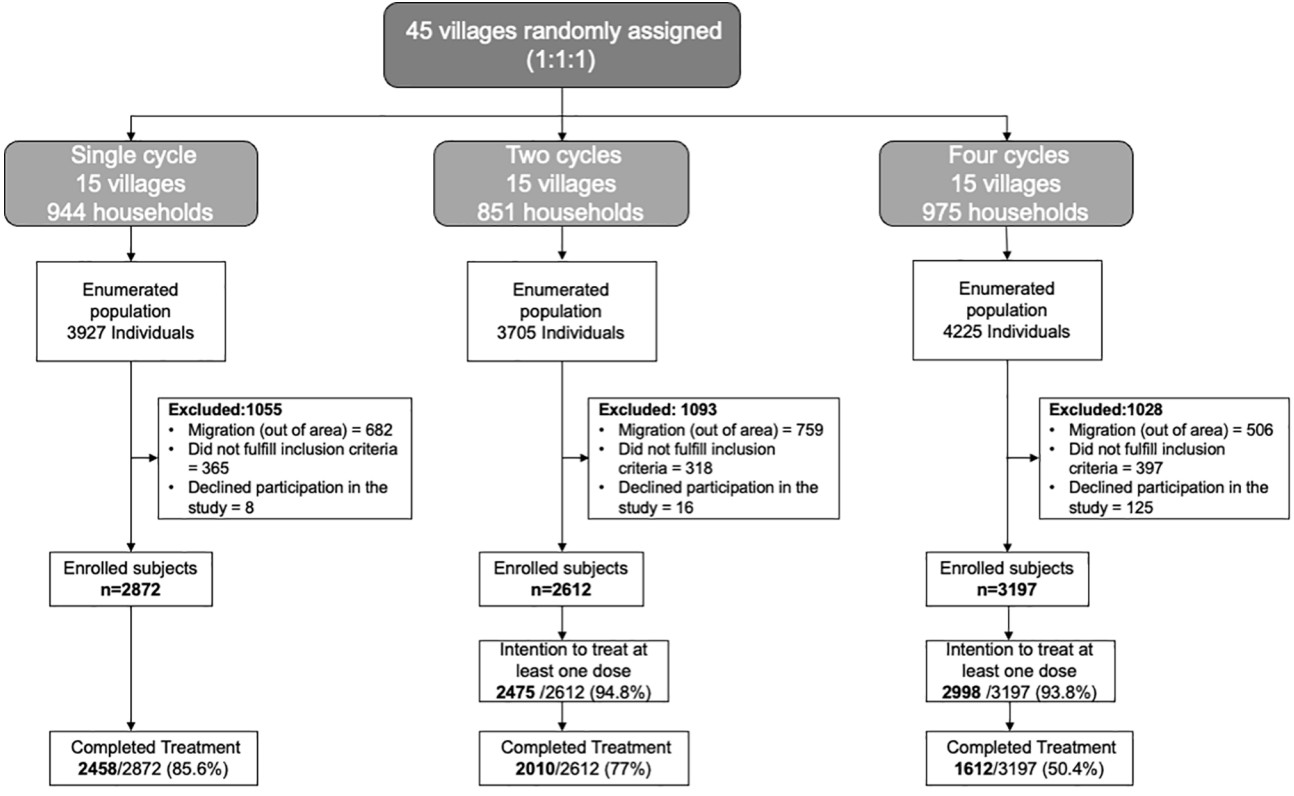

**Fig 1. Overview of participant enrollment and implementation of the mass drug administration (MDA) regimens.**

in the two-cycle group and 22.4% in the four-cycle group (Table 2). The mean (SE) EPGs were 52 (10), 144 (63) and 181 (39) in single, two-, and four-cycle groups, respectively.

In the multivariable analysis, male gender (aOR: 1.31, $P = 0.032$) and living in houses with mud flooring (aOR: 1.31, $P = 0.038$) were associated with an increased risk of hookworm infection; as was older age, with the highest risk in participants aged 45–54 years (adjusted odds ratio [aOR]: 5.26, $P < 0.001$). Drinking treated water was protective (aOR: 0.75, $P = 0.049$). Regular contact with animals was significantly associated with hookworm infection in the univariate (OR: 1.40, $P = 0.030$) but not in the multivariable analysis (aOR: 1.06, $P = 0.726$). Footwear usage, washing hands with soap and water after defecation, performing agriculture-related activities in the preceding six months, and socioeconomic status were not associated with hookworm infection risk (S2 Table).

### 3.3. Post-MDA prevalence and intensity *of* hookworm infection

After the first MDA cycle, the prevalence of hookworm infection decreased to 3.7% (95% CI, 2.6-5.1%), i.e., a relative reduction of 80% (72.4-85.9%) from the baseline prevalence of 18.5% [16], further decreasing to 1.8% (95% CI, 1.0-3.3%) after the second MDA cycle. Thereafter, the prevalence remained stable despite two additional MDA cycles after six months (1.3% [95% CI: 0.6-2.8%] and 1.6% [0.6-4.3%] after the third and fourth MDA cycles, respectively).

During the post-intervention period (3–24 months), the prevalence of hookworm infection ranged from 2.6-5.2% in the single-cycle group and from 4.0 to 7.9% in the two-cycle group. On the other hand, the prevalence of hookworm infection in the four-cycle group was constantly lower, ranging from 1.0-2.6% (Table 2). The post-intervention mean EPG of feces

**Table 2. Pre- and post-intervention prevalence and intensity of hookworm infection in the study villages assigned to the different MDA regimens.**

| Sample type | Single cycle | | Two cycles | | Four cycles | |
|---|---|---|---|---|---|---|
| | Prevalence (95% CI)* | Mean (SE) EPG¶ | Prevalence (95% CI)* | Mean (SE) EPG¶ | Prevalence (95% CI)* | Mean (SE) EPG¶ |
| Pre-intervention | 14.8% (11.1-19.5%) | 52 (10) | 18.5% (15.4%-22%) | 144 (63) | 22.4% (17.4-28.2%) | 181 (39) |
| 3-month post-intervention | 5.1% (3.1-8.2%) | 14 (4) | 5.7% (3.5-9.1%) | 17 (6) | 2.6% (1.5-4.6%) | 8 (5) |
| 6-month post-intervention | 5.2% (2.7-9.6%) | 12 (5) | 7.9% (5.3-11.5%) | 24 (7) | 1.8% (0.8-3.9%) | 2 (1) |
| 9-month post-intervention | 2.6% (1.1-5.9%) | 4 (3) | 5.0% (2.7-9.0%) | 14 (5) | 2.0% (0.9-4.0%) | 2 (1) |
| 12-month post-intervention | 5.0% (2.9-8.6%) | 12 (6) | 4.0% (1.8-8.4%) | 18 (9) | 1.2% (0.3-4.0%) | 3 (3) |
| 24-month post-intervention | 3.3% (1.4-7.7%) | 8 (4) | 4.7% (2.8-7.9%) | 13 (5) | 1.0% (0.03-3.1%) | 3 (1) |

* 95% CI adjusted for clustering at the village level.

¶EPG (eggs per gram of feces) counted by the McMaster technique; SE adjusted for clustering at the village level.

followed a similar pattern, ranging from 4-14 EPG in the single-cycle group, 13–18 in the two-cycle group and 2–8 in the four-cycle group (Table 2).

### 3.4. Assessment of MDA strategies

The post-MDA GEE analysis (adjusted for baseline prevalence/intensity of hookworm infection, age, and gender) showed no significant difference in the prevalence and intensity of hookworm infections between the single- and the two-cycle groups at 3-, 6-, 9-, 12- and 24-month post-intervention (Table 3). However, the prevalence of hookworm infection in the four-cycle group was significantly lower at 3-month (OR: 0.62, $P=0.024$), 6-month (OR: 0.41, $P=0.001$) and 12-month

**Table 3. Results of GEE analysis to ascertain the effect of the intervention (MDA) on hookworm prevalence and intensity.**

| Time post-intervention | Two cycles | | | | Four cycles | | | |
|---|---|---|---|---|---|---|---|---|
| | Prevalence* | | Intensity¶ | | Prevalence* | | Intensity¶ | |
| | OR (95% CI) ᐃ | P-value | IRR (95% CI) ᐃ | P-value | OR (95% CI) ᐃ | P-value | IRR (95% CI) ᐃ | P-value |
| 3-month post-intervention | 0.90 (0.49-1.63) | 0.727 | 0.82 (0.46-1.47) | 0.511 | **0.62 (0.41-0.94)** | **0.024** | **0.30 (0.20-0.45)** | **<0.001** |
| 6-month post-intervention | 1.34 (0.67-2.70) | 0.409 | 1.68 (0.78-3.62) | 0.189 | **0.41 (0.24-0.71)** | **0.001** | **0.19 (0.11-0.31)** | **<0.001** |
| 9-month post-intervention | 1.58 (0.66-3.82) | 0.305 | 2.12 (0.65-6.89) | 0.212 | 0.68 (0.41-1.15) | 0.149 | **0.36 (0.18-0.72)** | **0.004** |
| 12-month post-intervention | 0.60 (0.25-1.43) | 0.246 | 1.19 (0.39-3.64) | 0.754 | **0.41 (0.20-0.84)** | **0.016** | **0.29 (0.15-0.55)** | **<0.001** |
| 24-month post-intervention | 1.72 (0.74-4.01) | 0.210 | 0.85 (0.29-2.46) | 0.766 | 0.68 (0.39-1.17) | 0.161 | 0.52 (0.27-1.00) | 0.052 |

* Adjusted for baseline prevalence of hookworm infection, age and gender; fitted using binomial family and logit link function.

¶Adjusted for baseline intensity of hookworm, age and gender; fitted using negative binomial family and log link function.

ᐃ95% CI calculated using robust standard errors; adjusted for clustering at the village level.

(OR: $P=0.016$) post-intervention when compared to the single cycle group. Moreover, in the four-cycle group, the intensity of infection was significantly lower ($P<0.05$), as compared to the single-cycle group, at 3-, 6-, 9-, and 12-month post-intervention. There was, however, no statistically significant difference in the prevalence and intensity of infection between the three groups (one, two and four cycle) 24-month post-intervention (Table 3).

### 3.6. Adverse and serious adverse events

Overall, 21 adverse events (AE) were reported in 18 participants (three had two AEs each) during the study period. The commonly reported mild and moderate symptoms were gastritis (14/21), fever with rash or myalgia (2/21), urticaria (1/21), and giddiness (1/21). Three severe adverse events (SAE) were attributed to acute appendicitis, basal cell carcinoma of the left eye, and post-streptococcal glomerulonephritis with nephrotic syndrome, of which the latter succumbed to the illness. None of the SAEs were associated with the intervention.

## 4. Discussion

In this cluster-randomized trial, the hookworm infection rates were compared between a single cycle of community-wide MDA (albendazole) with two and four cycles, respectively, in an endemic tribal population in southern India. The intent was to determine whether multiple rounds of community-wide MDA optimally reduce hookworm burden in endemic communities and if the post-MDA gains sustain over time. Compared to the single-cycle group, the prevalence of hookworm infection in the four-cycle group was significantly lower at 12 months post-intervention, but this was not sustained at 24 months. Despite multiple rounds of community-wide MDA, albendazole was well tolerated with no SAE's or seizures reported among study participants, even though the study was conducted in a community with high prevalence of neuro-cysticercosis [28].

Evidence suggests that community-wide MDA significantly reduces the burden and morbidity of helminth infections, especially in hookworm-endemic areas [29], compared to a targeted approach. A drawback of the community-wide MDA, however, is bounce-back of infection prevalence to pre-treatment levels, even after repeated treatment, either due to rein-fection or persistence of infection in a subset of individuals [30], highlighting the potential role of environmental reservoirs in the transmission of STH. Hence, in this study, an additional cycle of community-wide MDA after one month (two MDA cycles), targeting potential reinfections from infective larvae in the environment, was hypothesized to be more effective than an annual community-wide MDA. In this study, however, the additional MDA cycle was not sufficient to reduce the prevalence and intensity of hookworm infection, possibly due to continued transmission from other sources, such as untreated humans [8,10] and animals [31,32], and poor WASH, as evidenced by the high rates of open defecation. Two additional cycles of MDA (one month apart) after 6 months (i.e., a total of four-MDA cycles), however, resulted in a sig-nificantly lower prevalence and intensity of hookworm infection up to one year post-treatment, which likely reflects the benefits of multiple rounds of MDA in endemic communities, although the gains were not visible after two years. Repeated deworming of infected individuals helps target and eliminate adult worms before they can reproduce and release eggs into the environment. Additionally, it ensures that those who missed previous MDA rounds or did not achieve complete clearance of infection receive adequate treatment. Together, these contribute to a sustained reduction in worm burden and environmental contamination with infective larvae, thereby lowering both the prevalence and the intensity of infection [33,34].

Bounce-back of infection [35] due to continuing transmission from untreated reservoirs could have been a potential reason for the observed lack of a statistically significant difference in the prevalence and intensity of hookworm infection between the single- and four-cycle groups two years after the last round of MDA. However, it is unlikely in this case as the prevalence of hookworm infection was similar in the four-cycle group 12- and 24-months post-intervention (Table 2). A more plausible explanation for this apparent lack of difference is the high statistical uncertainty and the low sensitivity of the microscopy techniques in low-intensity settings [36], which may have resulted in an inaccurate estimation of the

infection prevalence. Increasing the number of clusters or individuals to be tested [37] may have provided a more accurate estimation of the treatment effect.

The lower baseline hookworm prevalence observed in this study, compared to an earlier cross-sectional survey in the same tribal population [10], may be attributed to the selection of more accessible villages, anticipating adverse neurological events associated with asymptomatic neurocysticercosis in pork-consuming communities, such as the study population [19,20]. Villages with better road access generally have better access to health care and WASH facilities, resulting in lower hookworm prevalence [38,39]. The lesser number of stool samples tested per participant may also have resulted in underestimation of the true infection prevalence, as diurnal fluctuations in fecal egg counts can affect sensitivity of diagnostic tests in low endemic settings [40]. Testing multiple stool samples over consecutive days [36] or using a more sensitive diagnostic method, such as qPCR [41,42] could have provided more accurate estimates. Nonetheless, this is unlikely to have affected the study's overall conclusions as the randomization of the study villages would have helped mitigate potential misclassification bias [43].

In this study, the prevalence and intensity of hookworm infection increased with age, which is similar to what has been reported earlier [8,10,44], Males had a higher risk of infection, probably because of more prolonged contact with soil from agricultural activities, which is a known risk-factor of hookworm infection in this population [10]. Additionally, individuals living in a house with mud flooring were at a higher risk, probably because such flooring provides a favourable environment for the hookworm larvae to survive [45]. Surprisingly, however, footwear usage was not associated with hookworm infection in this study, possibly because of the widespread use of open-toed footwear (sandals, slippers) that fails to protect adequately against hookworm larvae [46].

The baseline prevalence of MHI infections in the study villages (<1.2%) was below the WHO target for STH elimination as a public health problem (<2% prevalence of MHI infections) [47]. Although this low baseline prevalence limits direct generalizability to higher-intensity communities, the findings of this study provide valuable insights for assessing the feasibility of STH transmission interruption in low-intensity settings. It is important to note that the WHO target of <2% prevalence of MHI infections is aimed at morbidity control rather than transmission interruption. It is estimated that interrupting STH transmission in endemic communities will require a post-treatment STH prevalence of ≤2% [48]. However, this threshold may be difficult to attain through targeted deworming of PSAC and SAC alone [49], particularly in hookworm-endemic areas where untreated reservoirs of infection sustain high reinfection rates [50]. Therefore, alternative deworming strategies need to be explored if transmission interruption is to be achieved.

This study has some limitations. The selection of villages located near main motorable roads may have introduced a selection bias. Additionally, seasonal migration outside the study area for employment, which was a major reason for non-participation, could have undermined the effectiveness of MDA in some of the study villages. Modeling studies have demonstrated that low annual seasonal migration rates can result in the re-introduction of STH infection, even in low-endemic settings [51]. There were differences in the baseline prevalence of hookworm between the intervention arms, although this is unlikely to have changed the overall conclusions as the highest prevalence was observed in the four-cycle group. It is also possible that the participants may have received deworming tablets from other sources, which may have resulted in more conservative estimate of the treatment effect.

## 5. Conclusion

This cluster-randomized community-intervention trial, conducted in a tribal population in southern India, evaluated the effectiveness of modified MDA strategies, which account for the extrinsic incubation period of hookworm, to identify the optimal deworming strategy to reduce hookworm burden in endemic populations. The results demonstrate that multiple rounds of community-wide MDA, targeting all age groups, were both safe and effective in reducing the prevalence and intensity of hookworm infection in endemic communities. Further research is needed to understand the frequency and duration of deworming required, and the need for other control measures such as improved sanitation and hygiene practices, to achieve long-term transmission interruption.

## Supporting information

**S1 Table. Age-specific (baseline) prevalence and intensity of hookworm infection.** * 95% CI adjusted for clustering at the village level. ¶ EPG (eggs per gram of faeces) counted by the McMaster technique; SE-standard error adjusted for clustering at the village level.
(DOCX)

**S2 Table. Results of logistic regression analysis to explore the factors associated with hookworm infection at baseline.** * 95% CI for ORs adjusted for clustering at the village level. ¶ Reference category: age group <5 years. § Reference category: SES category- low.
(DOCX)

**S1 Fig. Map of the study area showing the spatial distribution of study villages, by intervention group.** The map was created entirely from self-collected primary data for this study. The area boundaries, as well as all point locations, were digitized by the study team based on our own field data. No external basemap tiles, satellite images, proprietary map services (e.g., Google Maps, MapQuest), or third-party shapefiles (such as Natural Earth, OpenStreetMap, or other providers) were used to generate the underlying boundary layer shown in S1 Fig.
(TIF)

**S2 Fig. Demographic profile of (A) eligible and (B) enrolled participants; and participants providing stool samples for (C) baseline, (D) 3-month, (E) 6-month, (F) 9-month, (G) 12-month and (H) 24-month follow-up coprological surveys.**
(TIF)

## Acknowledgments

We are indebted to the study participants for their enthusiastic participation and support. We thank the Department of Health and Family Welfare, Government of Tamil Nadu, for permitting us to carry out this study in the tribal block. We also thank P.K. Saravanakumar and Rev. R. Raju for their help with field worker training and community mobilization. We acknowledge the support of the field workers, the laboratory, and the office staff of the Wellcome Trust Research Laboratory, Division of Gastrointestinal Sciences, Christian Medical College, Vellore.

## Author contributions

**Conceptualization:** Rajiv Sarkar, Vinohar Balraj, Vedantam Rajshekhar, Jayaprakash Muliyil, Simon J. Brooker, Roy M. Anderson, Gagandeep Kang, Sitara Swarna Rao Ajjampur.

**Data curation:** Vasanthakumar Velusamy, Srinivasan Venugopal.

**Formal analysis:** Rohan Michael Ramesh, Rajiv Sarkar, Vasanthakumar Velusamy, Srinivasan Venugopal.

**Funding acquisition:** Rajiv Sarkar, Kuryan George, Gagandeep Kang, Sitara Swarna Rao Ajjampur.

**Investigation:** Sitara Swarna Rao Ajjampur.

**Methodology:** Rajiv Sarkar, Anuradha Rose, Venkata R. Mohan, Vinohar Balraj, Vedantam Rajshekhar, Kuryan George, Jayaprakash Muliyil, Nicholas C. Grassly, Simon J. Brooker, Roy M. Anderson, Gagandeep Kang, Sitara Swarna Rao Ajjampur.

**Project administration:** Rajiv Sarkar, Vasanthakumar Velusamy, Srinivasan Venugopal, Anuradha Rose, Venkata R. Mohan.

**Supervision:** Vinohar Balraj, Jayaprakash Muliyil, Nicholas C. Grassly, Simon J. Brooker, Roy M. Anderson, Gagandeep Kang, Sitara Swarna Rao Ajjampur.

**Writing – original draft:** Rohan Michael Ramesh, Rajiv Sarkar.

**Writing – review & editing:** Vasanthakumar Velusamy, Srinivasan Venugopal, Anuradha Rose, Venkata R. Mohan, Vinohar Balraj, Vedantam Rajshekhar, Kuryan George, Jayaprakash Muliyil, Nicholas C. Grassly, Simon J. Brooker, Roy M. Anderson, Gagandeep Kang, Sitara Swarna Rao Ajjampur.

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
