## [Decision Letter · Decision Letter 0]

6 Nov 2025

PNTD-D-25-01140

Community-wide deworming strategies to reduce high hookworm burden in endemic communities: Results from a cluster randomized trial in southern India

Dear Dr. Ajjampur,

Thank you for submitting your manuscript to PLOS Neglected Tropical Diseases. After careful consideration, we feel that it has merit but does not fully meet PLOS Neglected Tropical Diseases's publication criteria as it currently stands. Therefore, we invite you to submit a revised version of the manuscript that addresses the points raised during the review process.

Please submit your revised manuscript within by Jan 05 2026 11:59PM. If you will need more time than this to complete your revisions, please reply to this message or contact the journal office at plosntds@plos.org. Please include the following items when submitting your revised manuscript:

We look forward to receiving your revised manuscript.

Kind regards,

Michael Cappello, M.D.

Academic Editor

Francesca Tamarozzi

Section Editor

Shaden Kamhawi

co-Editor-in-Chief

Paul Brindley

co-Editor-in-Chief

**Additional Editor Comments:**

The manuscript has been reviewed and I concur with the issues raised. Overall this is a well designed and implemented study. However, there are questions raised by the reviewer about the methods and results that when addressed will provide greater clarity. In addition, a revised manuscript should address the "overarching question" raised by the reviewer regarding the low prevalence of moderate to high intensity infections in the study population. The authors should explain how this affects interpretation of the findings of the study and implications for extrapolating results to other endemic communities.

**Journal Requirements:**

At this stage, the following Authors/Authors require contributions: Rohan Michael Ramesh, Rajiv Sarkar, Vasanthakumar Velusamy, Srinivasan Venugopal, Anuradha Rose, Venkata R. Mohan, Vinohar Balraj, Vedantam Rajshekhar, Kuryan George, Jayaprakash Muliyil, Nicholas C. Grassly, Simon J. Brooker, Roy M. Anderson, Gagandeep Kang, and Sitara Swarna Rao Ajjampur. Please ensure that the full contributions of each author are acknowledged in the "Add/Edit/Remove Authors" section of our submission form.

2) Tables should not be uploaded as individual files. Please remove these files and include the Tables in your manuscript file as editable, cell-based objects. For more information about how to format tables, see our guidelines:

https://journals.plos.org/plosntds/s/tables

**Reviewers' Comments:**

Reviewer's Responses to Questions

**Key Review Criteria Required for Acceptance?**

**Methods:**

-Are the objectives of the study clearly articulated with a clear testable hypothesis stated?

-Is the study design appropriate to address the stated objectives?

-Is the population clearly described and appropriate for the hypothesis being tested?

-Is the sample size sufficient to ensure adequate power to address the hypothesis being tested?

-Were correct statistical analysis used to support conclusions?

-Are there concerns about ethical or regulatory requirements being met?

Reviewer #1: Objectives are clear, study design is appropriate.

Details are needed on the selection and participation in the coprological survey. Please clarify how the random subset of participants were selected? Were they selected at the household level? At the community level? Any stratification by age? What proportion of the community were selected? Were participants from each community recruited proportional to population?

Please clarify the timing of the follow up in the three sets of treatment villages. Was the pre-intervention synchronized across the three treatments? And then three month post-intervention was staggered across the three treatments (i.e. 3 months after single cycle, three months after end of second cycle (one month after 1st treatment group), and three months after end of fourth cycle (8 months after the single cycle group)? Any seasonal impacts on transmission in the study communities?

**Results:**

-Does the analysis presented match the analysis plan?

-Are the results clearly and completely presented?

-Are the figures (Tables, Images) of sufficient quality for clarity?

Reviewer #1: Please provide demographic details on the participants in the coprological surveys across the timepoints. These could be included as a supplemental file; but important to see the characteristics of the participants on whom the prevalence and intensity data are based.

**Conclusions:**

-Are the conclusions supported by the data presented?

-Are the limitations of analysis clearly described?

-Do the authors discuss how these data can be helpful to advance our understanding of the topic under study?

-Is public health relevance addressed?

Reviewer #1: Discussion:

Line 354-357 – Results conclude no statistically significant difference between three different treatment groups 24 months after intervention (lines 318-320); Lines 354-357 assert significantly lower prevalence and intensity after four cycles. Please integrate the nuance here that is repeated in the next paragraph, of the difference no longer being apparent at 24 months.

Consider including the limitation that the criteria for WHO elimination as a public health problem were met prior to onset of the study.

**Editorial and Data Presentation Modifications?**

Reviewer #1: Minor issues:

Line 68 – ‘along-with’ doesn’t need a hyphen.

Line 75-76 – please clarify what the appropriate reference is. Findings from large-scale community intervention trials are referenced in citations 11 and 12, but 12 is a global modeling study, not a large-scale community intervention trial.

**Summary and General Comments:**

Reviewer #1: Community-wide deworming strategies to reduce high hookworm burden in endemic communities: Results from a cluster randomized trial in southern India

This study reports on an open-label, cluster-randomized community-intervention trial comparing community hookworm burden with a single cycle of albendazole MDA, two cycles one month apart, and four-cycles of single dose albendazole MDA (two cycles one month apart followed by two more cycles one month apart six months later). Authors randomly assigned 45 villages to one of the three treatments. Outcome was hookworm prevalence and mean intensity of infection at 12 months post-MDA.

Overarching question –

Given the emphasis on achieving and sustaining the WHO target for STH elimination as a public health problem (<2% prevalence of moderate-to-heavy intensity infections, line 418-419), it seems the study villages met that criteria at the onset of the study (line 262-263; <1.2% moderate to heavy intensity infections). How do you think that baseline condition affects the generalizability of your work to higher intensity communities?

PLOS authors have the option to publish the peer review history of their article (what does this mean?). If published, this will include your full peer review and any attached files.). If published, this will include your full peer review and any attached files.

.

Reviewer #1: No

**Figure resubmission:**
---

## [Editor Report · Decision Letter 1]

30 Mar 2026

Dear Professor Ajjampur,

We are pleased to inform you that your manuscript 'Community-wide deworming strategies to reduce high hookworm burden in endemic communities: Results from a cluster randomized trial in southern India' has been provisionally accepted for publication in PLOS Neglected Tropical Diseases.

Best regards,

Michael Cappello, M.D.

Academic Editor

Francesca Tamarozzi

Section Editor

Shaden Kamhawi

co-Editor-in-Chief

Paul Brindley

co-Editor-in-Chief

Thank you for responding to the comments on the original submission. The revised manuscript is much improved and suitable for publication.

---

## [Editor Report · Acceptance letter]

Dear Professor Ajjampur,

We are delighted to inform you that your manuscript, "Community-wide deworming strategies to reduce high hookworm burden in endemic communities: Results from a cluster randomized trial in southern India," has been formally accepted for publication in PLOS Neglected Tropical Diseases.

Best regards,

Shaden Kamhawi

co-Editor-in-Chief

Paul Brindley

co-Editor-in-Chief
